# Bee Stressors from an Immunological Perspective and Strategies to Improve Bee Health

**DOI:** 10.3390/vetsci9050199

**Published:** 2022-04-21

**Authors:** Hesham R. El-Seedi, Hanan R. Ahmed, Aida A. Abd El-Wahed, Aamer Saeed, Ahmed F. Algethami, Nour F. Attia, Zhiming Guo, Syed G. Musharraf, Alfi Khatib, Sultan M. Alsharif, Yahya Al Naggar, Shaden A. M. Khalifa, Kai Wang

**Affiliations:** 1Pharmacognosy Group, Department of Pharmaceutical Biosciences, Uppsala University, Biomedical Centre, P.O. Box 591, SE 751 24 Uppsala, Sweden; 2International Research Center for Food Nutrition and Safety, Jiangsu University, Zhenjiang 212013, China; 3International Joint Research Laboratory of Intelligent Agriculture and Agri-Products Processing (Jiangsu University), Jiangsu Education Department, Nanjing 210024, China; 4Department of Chemistry, Faculty of Science, Menoufia University, Shebin El-Kom 32512, Egypt; hanan.hmed557@gmail.com; 5Department of Bee Research, Plant Protection Research Institute, Agricultural Research Centre, Giza 12627, Egypt; aidaabd.elwahed@arc.sci.eg; 6Department of Chemistry, Quaid-I-Azam University, Islamabad 45320, Pakistan; asaeed@qau.edu.pk; 7Al nahal al jwal Foundation Saudi Arabia, P.O. Box 617, Al Jumum, Makkah 21926, Saudi Arabia; ahmed@alnahalaljwal.com.sa; 8Chemistry Division, National Institute of Standards, 136, Giza 12211, Egypt; nour.fathi@nis.sci.eg; 9School of Food and Biological Engineering, Jiangsu University, Zhenjiang 212013, China; guozhiming@ujs.edu.cn; 10H.E.J. Research Institute of Chemistry, International Center for Chemical and Biological Sciences, University of Karachi, Karachi 75270, Pakistan; musharraf@iccs.edu; 11Department of Pharmaceutical Chemistry, Kulliyyah of Pharmacy, International Islamic Univetsity Malaysia, Kuantan 25200, Malaysia; alfikhatib@iium.edu.my; 12Faculty of Pharmacy, Universitas Airlangga, Surabaya 60155, Indonesia; 13Biology Department, Faculty of Science, Taibah University, Al Madinah 887, Saudi Arabia; ssharif@taibahu.edu.sa; 14Zoology Department, Faculty of Science, Tanta University, Tanta 31527, Egypt; yehia.elnagar@science.tanta.edu.eg; 15General Zoology, Institute for Biology, Martin Luther University Halle-Wittenberg, Hoher Weg 8, 06120 Halle, Germany; 16Department of Molecular Biosciences, The Wenner-Gren Institute, Stockholm University, SE 106 91 Stockholm, Sweden; shaden.khalifa.2014@gmail.com; 17Institute of Apicultural Research, Chinese Academy of Agricultural Sciences, Beijing 100093, China

**Keywords:** honeybees, immunity, agrochemicals, nutrition, ecological stressors, sustainable beekeeping

## Abstract

Honeybees are the most prevalent insect pollinator species; they pollinate a wide range of crops. Colony collapse disorder (CCD), which is caused by a variety of biotic and abiotic factors, incurs high economic/ecological loss. Despite extensive research to identify and study the various ecological stressors such as microbial infections, exposure to pesticides, loss of habitat, and improper beekeeping practices that are claimed to cause these declines, the deep understanding of the observed losses of these important insects is still missing. Honeybees have an innate immune system, which includes physical barriers and cellular and humeral responses to defend against pathogens and parasites. Exposure to various stressors may affect this system and the health of individual bees and colonies. This review summarizes and discusses the composition of the honeybee immune system and the consequences of exposure to stressors, individually or in combinations, on honeybee immune competence. In addition, we discuss the relationship between bee nutrition and immunity. Nutrition and phytochemicals were highlighted as the factors with a high impact on honeybee immunity.

## 1. Introduction

The European honeybees (*Apis mellifera* L.) is considered one of the most important agricultural pollinators worldwide. They play a key role in food productivity by pollinating various plants [1,2]. One-third of a person’s diet comes from insect-pollinated plants, and honeybees are responsible for the pollination of over 80% of flowering plants. Without honeybees pollination, crop yields would decrease by >90% [3]. Therefore, there is major international concern related to bee colony loss [4,5]. Overall, 52 of the 115 leading world food commodities rely on honeybees. Thus, in the honeybees loss scenario, fruit quality or quantity yields would be reduced for 16 commodities (by 90–40%), modestly reduced (10–40%) for 19 commodities, and somewhat reduced (by 10%) for 13 commodities [3]. Pollination fees for blooming cherries, plums, and almonds have increased. For example, almond pollination fees have increased by ~180%; this increase mainly occurred between 2004 and 2006. Previous study estimated that a 10% rise in the bees colony’s winter death rate results in 16% drop in overall almond pollination revenue [6].

Colony collapse disorder (CCD), which first emerged in the US in 2006, caused huge colony losses and posed challenges for crop pollination, which is the major service of the apicultural industry in North America [7]. The observed losses between 1961 and 2007 recorded in Europe and North America were 26.5% and 49.5%, respectively. Honeybee colonies have increased, primarily in Asia (426%), Africa (130%), South America (86%), and Oceania (39%) [8]. The following factors have been implicated in honeybee losses in different parts of the world: honeybee diseases, parasites, in-hive chemical substances, agrochemicals, genetically modified (GM) plants, modified land-use, changed and alteration in the cultural practices, beekeeping practices, as well as the climate change [9,10,11,12,13,14].

Although most scientists understand the multiple origins of bee colony losses, a full understanding of the mechanistic basis of this devastating occurrence is required. What is the common trend in colony losses that can be linked to diverse stress factors depending on the circumstances? How can bees respond to a comprehensive environmental challenge with a coordinated stress response against biotic and abiotic stress agents, which frequently have a synergistic effect and result in a considerably stronger negative impact of parasites and pathogens (often coexisting in complicated associations)?. These are the most pressing issues that must be addressed [15].

## 2. Honeybee Immunity

The widespread agreement on the multifactorial origins of colony collapse and its frequent correlation with high pathogen and parasite loads indicate that the immune system is the most targeted, and its activity can be altered by a variety of stressors [15]. Bees have an innate immune system, which includes physical barriers, cellular, and humoral responses to defend against pathogens and parasites. The physical barrier includes an exoskeleton cuticle and the peritrophic membranes lining the digestive tract to prevent the entry of pathogenic organisms into the body cavity [16]. The recognition of pathogen-associated molecular patterns by recognition receptors triggers the innate immune system [17]. As a result, hemocytes that represent the primary mediators of cellular immunity will be activated, including phagocytosis, nodule formation, encapsulation, as well as the initiation of phenol-oxidase that regulates coagulation or melanization, or the synthesis of antimicrobial peptides (AMP), such as abaecin, apidaecin, hymenoptaecin, and defensin [18]. The immune system of honeybees possesses orthologues for the major members of immune pathways comprising the following: Toll (transmembrane signal transduction pathway), immune deficiency (Imd), Jun-N-terminal kinase, and JAK/STAT (Janus kinase/signal transducers and activators of transcription). RNA interference (RNAi), known as RNA silencing, is an important antiviral defense mechanism in insects, including honeybees. The efficacy of RNAi-mediated treatment against honeybee viruses and the fact that honeybee viruses encode for putative virus-encoded suppressors of RNAi shows that RNAi is an important honeybee antiviral defense mechanism [10,19]. In addition, several studies implicate the involvement of innate immune pathways (Jak-STAT, Toll, and Imd) and non-sequence-specific dsRNA-mediated antiviral defense as part of the immune responses in honeybee [20].

Bees are creatures that exhibit social immunity to prevent parasite infection from spreading among colony members. Honeybee workers use hygienic practices to remove diseased brood [21]. Furthermore, social fever develops when bees cooperate to raise the temperature within the colony to resist the heat-sensitive fungal disease *Ascosphaera apis*, known as chalkbrood disease [22]. Grooming, which is the physical removal of parasitic mites from the bodies of adult bees by individual workers or their nest-mates, is one of the most important defense modes against the ectoparasitic mite *Varroa destructor* [23]. Other mechanisms of social immunity such as propolis collection is used for nest constriction. It helps in declining investment in the immune response of 7-day-old bees and enriches the health and productivity of the colony [24]. Previous study showed that a propolis envelope reduced the clinical symptoms of American foulbrood (AFB) two months after the challenge, compared with those of colonies without a propolis envelope. Additionally, it protected the brood from pathogenic infection [25]. Glucose oxidase (GOX) is an antiseptic enzyme found in nectar and larval diet, and an additive to prolong the products’ shelf life. In the hypopharyngeal glands, GOX is a catalytic enzyme that catalyzes the conversion of β- d- glucose to gluconic acid and hydrogen peroxide (H_2_O_2_) [26] and provides the social immunity. H_2_O_2_ functions as an antiseptic, preventing pathogen growth in honeybee larval diet [27].

Owing to the unique reactivity of the immune system to foreign bodies, it provides safety and guarantees the survival of honeybees. The immune system must be strengthened to prevent and control pathogenic viral infections [28]. The maintenance and repair of the immune system are the most costly physiological processes [29]. It has been discovered that nutrition has a significant impact on immunological response [30]. In honeybees, it depends on the type, quality, and diversity of nutrition [31]. It is important to minimize bee stress by (1) providing floral-rich vegetation, (2) minimizing pesticide usage by introducing more organic farming practices, (3) improving existing quarantine measures on bee movements, and (4) effectively monitoring honeybee populations for the development of future management strategies [32]. Herein, we summarized and discussed the effects of different stressors on honeybee immunocompetence, either individually or in combination. Furthermore, we discussed the relationship between nutrition and bee immunity, as nutrition and phytochemicals were discovered to be among the most influential strategies for improving honeybee immunity [33].

## 3. Main Causes of Honeybee Colony Losses

### 3.1. Varroa Mite

The ectoparasitic mite *V. destructor* is considered one of the most important factors behind the recent high annual loss of honeybee colonies. The mite directly damages bees by feeding primarily on honeybee fat body tissue and not hemolymph [34,35]. It has different impacts in different parts of the world, with increased morbidity and colony losses in honeybee colonies in North America, Europe, and Asia. Although this mite has been prevalent in Brazil for many years, no instances of colony losses among Africanized honeybees have been reported [36,37,38].

The physical damage caused by the mite has been reported to suppress the bees’ immune response [39]. Varroa parasitism has inhibited the expression of genes encoding immunity (hymenoptaecin and defensin), longevity, and stem cell proliferation in honeybees [40].

The mite is unlikely to cause the collapse of hives; however, it acts as a vector for a cocktail of viral disease agents, which are among the probable causes for CCD, including the deformed wing virus (DWV), kashmir bee virus, sacbrood virus (SBV), acute bee paralysis virus, and Israeli acute paralysis virus (IAPV) [41,42,43,44].

### 3.2. Nosema spp.

Nosemosis is a disease that affects honeybees and is caused by intracellular parasites (*Nosema apis* and *Nosema ceranae*) that infect the adults’ midgut epithelial cells [45]. *N. ceranae* infection is highly pathogenic for honeybee colonies, significantly reducing the colony size, brood rearing, and honey production, and increasing winter mortality. In persistent infections, the pathogen can impact colony performance by reducing a colony’s ability to regulate hive temperature or by killing the entire colony [46]. For the European honeybee, *N. ceranae* reduced homing and orientation skills, and altered the metabolism of forager bees [47].

*N. ceranae*-infected bees showed changes in the level of ethyl oleate (EO), which is, to date, the only primer pheromone identified in workers that is involved in foraging behavior. Infected bees with high EO titers have a short lifespan [48]. The immune response of bees against *N. ceranae* comprises the early initiation of AMPs via the Toll and Imd pathways, and the immune response is hampered after seven days of infection [49].

### 3.3. Viral Pathogens

For honeybees worldwide, over 24 viruses have been discovered, some of which can have major health repercussions [50]. A highly prevalent and relatively virulent virus transmitted by *V. destructor* which impacts the health of honeybee colonies worldwide is DWV [11]. DWV-induced honeybee loss, coupled with a long-term decline in beekeeping, has become a serious threat to the adequate provision of pollination services, which threatens food security and ecosystem stability [51].

Nitric oxide synthase is produced in high levels of DWV and leads to the formation of interferon-g (IFN-g), which has an inflammatory impact [52]. The Varroa mite can disrupt the dynamics of DWV within the host, turning a cryptic and vertically transmitted virus into a rapidly replicating killer that reaches deadly levels late in the season. The down-regulation of the transcription factor, nuclear factor kappa-B (NF-kB), gene family, is linked to the destabilization of DWV infection [41].

In colonies infected with DWV, *V. destructor* impacts two immune genes, relish and defensin. Relish up-regulation and defensin down-regulation expression were closely linked to high DWV and mite loads [53]. Additionally, the DWV titer was linked negatively to encapsulation and melanization responses [54]. Contrarily, high titers of DWV alone did not affect the expression of the AMPs and the genes involved in the regulation of melanization [55]. By limiting vitellogenin (Vg) expression, DWV had a favorable impact on the titer, and a negative relationship was observed between the Vg levels and Varroa mite infestation levels [44,56]. Vg is important for bees behavior and development and is linked to energy homeostasis [57].

SBV is a picornavirus in the genus Iflavirus which causes the failure of bee larva to pupate, resulting in the bee’s death [58]. Furthermore, SBV was detected in the hemolymph, ovary, and abdominal shell throughout the growth stages of workers and queens in *A. mellifera* colonies [59]. It caused the up-regulated expression of AMPs and down-regulated expression of the prophenoloxidase-activating enzyme (PPAE). Additionally, the up-regulation of the expression of putative serpin suppressed the melanization pathway [55].

Israeli acute paralysis virus (IAPV) has been identified as a major cause of CCD [60]; although other studies did not directly associate CCD and the presence of IAPV, successive replication of IAPV was shown to affect the colonies health and thus possibly determine their survival [61,62,63]. In addition, IAPV is a source of infection not only to honeybees, but also bumblebees [64], stingless bees [65], hornets [66], and even ants [67]. IAPV is an RNA virus of honeybees linked to colony losses [68]. IAPV caused immunological signaling abnormalities in adult bees that were more pronounced than in brood. Endocytosis genes, such as Cbl, which is a signal transducer and activator of transcription (STAT), protein inhibitor of activated signal transducer, and hopscotch were up-regulated [68].

### 3.4. Pesticides

The exposure of honeybees to pesticides compromises their immune responses, navigation ability, learning, and memory [69]. Pesticides in sub-lethal quantities can be harmful to honeybees, as it may not kill bees but reduce their performance and survival during foraging. Bees exposed to pesticides had high vulnerability to infections and hence became means to spread diseases to other parts of the colony or other colonies via the shared use of flowers [9].

Thiamethoxam (TMX), a kind of neonicotinoid, does not promote collapse in healthy colonies in spring. This does not rule out the possibility that colonies will be more vulnerable later in the year, manifested when their capacity to replace lost workers diminishes gradually [70]. In bees infected with Varroa mites, TMX altered the gene expression of their immune systems [71]. Similar changes in immune genes were observed in bees exposed to imidacloprid and infected with Varroa where the potent effect is variation over the three developmental stages (white-eyed pupae, brown-eyed pupae, and newly emerged bees) [72]. The exposure to imidacloprid decreased AMPs in the hemolymph and encapsulation response, owing to the reduction in hemocytes and prophenoloxidase [73].

On the other hand, other studies debated the above notion, demonstrating that neonicotinoids did not negatively influence honeybee colonies and that the honeybee was relatively resistant to neonicotinoid effects in real-world scenarios [74]. These claims were supported by field studies that demonstrated great adaptability of honeybee colonies to environmental challenges, especially when using a sub-lethal dose of neonicotinoids. Equally interesting, clothianidin (CLO) treatment resulted in an increased colony power (the number of brood and adult bees) and a decrease in mortality [74].

Taken together, it was thought that exposure of bees to more than one pesticide can produce a synergistic, antagonistic, or additive effect. Exposure to more than one pesticide agrochemical including pesticides, insecticides, herbicides, and fungicides during field-realistic interaction was synergistically harmful [75].

### 3.5. Malnutrition

Food resources are collected in large quantities by bee colonies to prepare for scarcity and are stored as honey and bee bread. This is because nutrition deficiency and deterioration can affect development and bee lifespan and increase the likelihood of infestation by a parasite, virus, or disease, resulting in honeybee mortality [76,77]. Bees fed only on water and sugar (low protein diet) exhibited higher mortality and viral load rates, compared with bees fed on higher pollen diets. These results showed that poor nutrition can suppress immunity and that a different host’s nutrition can alter specific components of the immune system [78]. Supplementary nutrition such as homemade sugar syrups can cause undesirable effects on bee health due to the presence of toxic compounds, such as hydroxylmethylfurfural (HMF). In addition, the preparation conditions of these sugar syrups such as temperature or addition of an acidifying substance resulted in a higher amount of HMF and bee mortality [79].

The other threat that may generally affect honeybees health and immune system specificity is GM crops [80]. GM crops are used to produce desired traits, particularly insect and herbicide resistance [81]. GM crops may affect honeybees directly and indirectly. Indirectly, by interfering with volatile chemicals that are efficient for attracting honeybees. The indirect effect is the variations in the quantity and consistency of plant secondary compounds that may cause a deficiency in the main nutrition, which is related to immune system competence [29,33]. Through nutrition, GM can directly suppress insect digestion enzymes, such as amylase and proteases [82]. Accurate data on the direct effect are inadequate, but the fear of this effect exists [83,84]. The available experiments are only performed in the laboratory under artificial conditions, and the observations are based on an unnatural situation. Thus, experiments under natural conditions are greatly required [85].

### 3.6. Other Causes

Heavy metal pollution originates from various sources. Heavy metals may be ingested by honeybees through water sources and by foraging for nectar and pollen from plants that have already stored heavy metals [86]. Cd, Pb, As, Hg, Ni, and Cr are particularly harmful to living beings due to their high toxicity. Furthermore, elements such as Cu, Fe, Se, Zn, Mn, and Co, which are required for a range of biochemical and physiological processes, can be hazardous to bee colonies [87]. Cd pollution caused significant cellular damage in fat bodies, which was considered to be the cause of a decreased ability to eliminate pathogens [88]. Honeybee microbiomes and metabolomes were impacted [89]. Co, Mn, and Cu presence at xenobiotic levels lowered the phagocytic index and altered the profile of bees’ low-molecular weight proteins [90].

Urbanization is one of the biggest challenges to wild plants and pollinators, including bees [91]. Heavy metals such as Pb and Cd are found in high concentrations in urban areas. In urban areas, bees were more susceptible to disease transmission, particularly the black queen cell virus (BQCV) and fungal pathogen *N. ceranae*. This effect did not appear to be mediated by immunity, as determined by immune-gene expression, which was not affected by urbanization [90]. Moreover, there was no evidence of immunocompetence differences between managed and feral bees [92].

Nanoparticles (NPs), which are used in a variety of industrial applications, negatively impact human and animal health, including honeybees. CdO or PbO NPs produced histological and cellular abnormalities in honeybee workers’ midgut epithelial cells [93]. When exposed to CdO or PbO NPs, separately or in combination, acetylcholinesterase (AChE) activity and the expression of a variety of stress-related detoxifying enzymes were inhibited. Furthermore, the rate of feeding and survival reduced [94].

The rapid expansion of the telecommunications industry has resulted in a massive increase in the number of mobile phones and the rapid deployment of cell towers across the globe. According to certain studies, honeybees do not rely on the electromagnetic field (EMF) to navigate, and many apiaries that are experiencing losses are in rural areas where cell phone service is absent. The World Health Organization confirmed the same data; however, some researchers revealed that there was standard evidence that the EMF could cause damage in honeybees [95,96]. It was associated with increased bees activity, increased inside temperature, increased queen loss, abnormal real cell production, weight loss, and reduced operculated brood [97]. Chronic radiofrequency EMF exposure significantly reduced the hatching of honeybee queens [98]. In addition, activities related to transport, and artificial water surfaces were thought to be associated with high honeybee colony losses [99].

One proposed reason for honeybee losses was improper beekeeping practices, and it was reported that amateur beekeepers with small beehives and no experience in beekeeping had twice the winter mortality rate than experienced beekeepers [38]. Beekeepers should maintain strong colonies with a young and healthy queen and control the level of Varroa mites by accurately timed and effective control measures. In addition, long-term beekeeping experience is beneficial in reducing winter losses. Mite management should be a top focus for reducing colony losses. Multiyear surveys and surveillance studies are necessary to solve these questions and detect emerging problems [38]. Beekeeper training is warranted to encourage good beekeeping practices and to detect the clinical signs of disease early [12].

## 4. Interaction between Different Stressors Affects the Bees Immunocompetence

### 4.1. Interaction between Pesticides and Pathogens 

As previously mentioned, many factors including pesticides, diseases, and malnutrition lead to bees’ decline in different regions worldwide. These threats are frequently interconnected, and it is unlikely that colony losses are caused by single stressors (Figure 1, Table 1 and Table 2) [100].

Interactions between pesticides and pathogens may play a role in increased honeybee colony losses, including CCD and other pollinator reductions worldwide [101]. CLO inhibits NF-κB immune signaling in insects at sub-lethal dosages, and CLO and imidacloprid compromise honeybees antiviral defenses regulated by transcription factors [102]. Although other studies estimated that increased pesticides may not always result in increased viral loads [11], exposure to neonicotinoid pesticide imidacloprid, in the presence of the gut parasite *N. ceranae*, increased the levels of enzymes such as catalase (CAT) and glutathione-S-transferase in the heads of bees. These enzymes are involved in pesticide and parasite resistance to xenobiotics and oxidative stress. Furthermore, stressors affected midgut enzymes, i.e., carboxylesterase alpha (CaE) and carboxylesterase para (CaE p), which are engaged in metabolic and detoxifying processes [101,103,104,105]. In honeybees, the interactions between thiacloprid and *N. ceranae* caused Nosema to increase, regardless of the thiacloprid dosage [105]. Al Naggar and Bear conducted a series of experiments in which honeybee workers were exposed to sub-lethal field-relevant concentrations of a novel pesticide (Flupyradifurone) for a brief period during their larval development and early adulthood. They tested the long-term effects of a single short exposure to the pesticide and a second ecological stressor later in life in the absence and the presence of the pesticide. They concluded that brief exposure to sub-lethal pesticides during development or early life was sufficient to induce effects later in life, and this could cause a decline in bees health [9].

Imidacloprid and *V. destructor* played important roles in honeybees population and individual bee survival decline, owing to the inhibition of Vg titer, which modulates bees growth and behavior [57]. When bees were exposed to chronic bee paralysis virus (CBPV) at high imidacloprid doses, the viral load and mortality increased. Low dosages did not influence the viral titer [106]. The effect of TMX on *V. destructor*-infected bees changed the gene expression in the immune system. Spaetzle, AMPs, abaecin, and defensin-1 were all down-regulated in white infected pupas, while lysozyme-2 was up-regulated. By adding TMX to the Varroa-infested group, the expression of spaetzle, hopscotch, basket, and polyphenol oxidase (PPO) genes was significantly elevated in brown-eyed pupae. Oppositely, TMX did not affect defensin-2 in white- and brown-eyed infested pupae [71].

### 4.2. Interaction between Pesticides and Poor Nutrition

The combination of low diet and chemical exposure affects bee survival synergistically (−50%). The interaction reduced food consumption (−48%), hemolymph levels of glucose (−60%), and trehalose (−27%) [107]. Researchers have indicated that various insecticides indirectly influence honeybees’ health via diet by suppressing immunity-related genes and negatively altering NFB immunological signaling. It can significantly impair the honeybees’ immune system, reducing the bees antiviral defense regulated by this transcription factor. This may cause direct death or become easy prey to predators [102,108,109]. Bees survival was reduced as a result of the interaction of restricted nectar and nectar availability with neonicotinoid exposures, such as CLO and TMX [107]. Another study estimated the impact of flupyradifurone (FPF) exposure (regarded as bee-friendly, according to standard risk assessment) on poor nutrition where it reduced bee survival, food consumption, flight success, and thermoregulation of honeybees [110].

### 4.3. Interaction between Pathogens and Poor Nutrition

Nutritional deficiencies increased pathogen load and reduced adult longevity and survival [111]. Diseases altered foraging behavior by reducing foraging abilities or altering floral preferences. In addition, pollination services were impaired when bee populations were reduced or feeding habits were altered [112]. The interaction between poor nutrition and pathogenic infection increased the rate of colony mortality and reduced the ability of bees to fight other stressors [113]. Disease infection (viral, fungal, and bacterial pathogens), reduced bee nutrition (caloric needs, dietary requirements, nutrient storage, gut physiology, and microbiota), and made bees more susceptible to diseases and vice versa [33].

### 4.4. Interaction between Parasites and Pathogens

The interactions between the DWV and ectoparasitic mite, *V. destructor*, resulted in DWV replication and increased Varroa reproductive output [114]. *V. destructor* can vector IAPV in honeybees and are capable of IAPV replication. The density of Varroa mites and the duration of exposure to the mites were positively related to the copy number of IAPV in bees. Furthermore, the mite–virus association may reduce the host immunity, promoting high levels of virus replication. Varroa mites provide a plausible route for IAPV transmission in the field and may significantly contribute to the honeybee diseases associated with CCD [34,62].

**Table 1 vetsci-09-00199-t001:** Summary of laboratory studies discussing the interaction between stressors on honeybees (*Apis mellifera* L.).

First Stressor	Second Stressor	Model	Results	References
*Varroa destructor*	Neonicotinoid insecticides	Honeybees	Reduction in survival of long-lived winter honeybees	[115]
	Deformed wing virus (DWV)	White-eye honeybees pupae	Higher virus replication	[116]
	Poor nutrition	Freshly emerged honeybee workers	Reduction in body weight Reduction in abdominal protein level Increased head to abdomen protein ratio	[117]
*Nosema ceranae*	Insecticide fipronil	Emerging honeybees	Decreased survival	[118]
	DWV	Newly emerged honeybee workers	Higher virus replication in infected bees in a dose- and nutrition-dependent manner	[119]
	Neurotoxic insecticides	Emerging honeybees	Increased mortality of immunity-related genes Strong alteration of midgut immunity	[104]
	*Thiacloprid*	Freshly emerged honeybee workers	Higher Nosema replication	[105]
*Microsporidia Nosema*	Neonicotinoid (imidacloprid)	Young Africanized honeybees	Higher mortality rate Higher pathogen load	[120]
Thiamethoxam	*N. ceranae*	Larvae and adult honeybees	Gene expression patterns change with time in each treatment	[121]
	Chronic bee paralysis virus (CBPV)	Nine-day-old bees	Higher viral titers Increased mortality	[122]
	CBPV	Emerging honeybees	High pesticides increased mortality without an increase in viral titers Low pesticide has no effect on mortality, however, increases in viral titers	[123]
Imidacloprid	*N. ceranae*	Honeybee queens	Decreased metabolic and detoxification functions Increased mortality Reduced lifespan	[103]
	*N. ceranae*	Honeybees	Higher Nosema infection load Increased mortality of honey bee colonies	[101]
Thiacloprid	Microsporidian *N. ceranae* and BQCV	Larval and adult honeybees	Elevation of viral load Increased individual mortality	[124]
Clothianidin	Nosema spp.	Emerging honeybees	No any synergistic effect	[125]
Fipronil and Thiacloprid	*N. ceranae*	Emerging honeybees	Higher Nosema infection Higher mortality	[126]
Poor nutrition	Israeli acute paralysis virus (IAPV).	Honeybees	Elevation of bee mortality	[113]

**Table 2 vetsci-09-00199-t002:** Summary of field studies discussing the interaction between stressors on honeybees (*Apis mellifera* L.).

First Stressor	Second Stressor	Model	Results	References
*Varroa destructor*	Clothianidin	Honeybees	Reduced weight and number Up-regulated differentially expressed genes (DEGs) associated with metabolism	[127]
	Deformed wing virus (DWV)	Africanized honeybees	Inhibition of immunity Reduction in lifespan	[128]
	Neonicotinoid insecticide imidacloprid	Honeybees	Reduce homing success of foragers High *V. destructor* load	[129]
	Poor nutrition	Freshly emerged honeybee workers	Death of bee about 40% Ageing enhancement	[117]
	Viruses (Acute-Kashmir-Israeli and DWV, acaricide)	Honeybees	Higher virus replication immune-suppression	[130]
	DWV	Honeybees	Down-regulation of a member of the NF-Κ*β* Increase in bees mortality	[41]
Nosema	Herbicide glyphosate and the fungicide difenoconazole	Emerging honeybees	Induces strong physiological disturbances Reduced honeybees longevity	[131]
*Nosema apis*	Neonicotinoid pesticide Thiamethoxam	Honeybee workers	Increase in mortality Reduction in immunocompetence	[132]
*V.* *destructor*, Nosema spp.	Imidacloprid	Honeybees	Reduced the flight performance by ~24% No effect on colony size	[133]
Acaricides	*N. ceranae*	Newly emerged honeybees	Reduction in ethyl oleate as primer pheromone	[134]
Parasites	Pesticides	Honeybees	Increased mortality	[135]
Imidacloprid	CBPV	Adult honeybee workers	Higher virus load and mortality in case of higher doses Lower dosages had no influence on virus titer	[106]
Clothianidin	Varroa	Honeybees	No elevation of immune gene expression	[74]
	DWV	Honeybees	Decreased immunocompetence expression Higher replication of the DWV	[102]
	Pathogens (RNA viruses, DNA virus, Nosema SP, and beneficial bacteria)	Bumblebees	No increase in viral titers Impairs reproduction of queens and males	[136]
	Pathogens	Honeybees	No adverse effect on honey bee colonies No increase in pathogens titers	[137]
	Infection with parasites and diseases	Honeybees	No increase in titers of several viruses No impact on success of colony or honey yield	[138]
RoundupVR	*Nosema microsporidia*	Emerged and adult honeybees	Reduced survival rate and increased food consumption of the bees	[139]
Thiamethoxam	*N. ceranae*	*Apis mellifera carnica*	Epithelium degeneration Higher mortality	[140]
	*N. ceranae*	Larvae and adult honeybees	Highest mortality rate Decreased immunocompetence expression	[121]
	DWV	Newly emerged bees	The chance of not returning to the hive after the first flight was raised Decreased survival	[141]
	Chronic bee paralysis virus (CBPV)	Emerging honeybees	High pesticides increased mortality without an increase in viral titers Low pesticide has no effect on mortality but increases in viral titers	[123]
	CBPV	Adult honeybee workers	Higher viral titers Increased mortality	[122]
Neonicotinoid insecticides	*V. destructor*	Newly emerged bees	Decreased immunocompetence expression	[142]
Insecticide Flupyradifurone (FPF, Sivanto^®^)	Nutritional stress	Foraging honeybees	Reduced bee survival and food consumption	[110]
Pesticides (fipronil, thiamethoxam and boscalid)	*N. ceranae*	Newly emerged honeybees	Gut microbiota dysbiosis	[143]
Neonicotinoid pesticides	Nutritional stress	Honeybees	Reduction in bee survival Reduction in food consumption	[107]
Pesticides (dimethoate, clothianidin and fluvalinate)	American foulbrood (AFB)	Honeybees	Higher mortality Reduction in hemocyte counts	[144]
Poor nutrition	Virus infection	Honeybees	Elevation of bee mortality	[113]
	Thiamethoxam	Bumblebees micro-colonies	Slower growth Reduced reproductive efforts	[145]
	Thiamethoxam	*Apis mellifera ligustica*	Negative impact on hypopharyngeal gland development	[146]

## 5. Strategies to Enhance Honeybee Immunity

### 5.1. Fortified Nutrients

Currently, honeybees face numerous threats that hinder their survival. Special attention should be paid to beekeepers, bee supplements, and nutrition to limit the risk of viruses, other diseases, and agrochemicals. Beekeepers should support colonies with suitable supplementary feeding during dearth periods [147]. In addition, the diversity of nutrition that comes from different natural plant sources can improve honeybee immunity, antiviral, and antimicrobial properties [33]. A diet with pollen from different plants or high-quality single pollen organs can enhance honeybees’ immunity and survival as shown in Figure 2.

Using a mixed pollen diet of three pollen species indirectly increased bee larvae immunocompetence and resistance to fungal diseases, such as *Aspergillus fumigatus* [111]. Experiments on two different groups of honeybees correlated with these findings. The first group exclusively ate clover pollen, while the second ate clover and partridge pea (*Chamaecrista fasciculate*) pollen. In the absence of the pathogen, they exhibited no change in mortality, however, when the virus was administered, they showed a significant difference. When honeybees were infected by a virus, a combination of clover and partridge pea significantly reduced mortality and enhanced immunity and viral resistance [148]. Di Pasquale estimated that young nurse bees were affected by pollen quality, not multiflora pollen [31]

Furthermore, protein nutrition is the key factor to improve the immunity of honeybees and support the survival of honeybee communities [141]. For Nosema infection, the availability of various floral resources can increase infection tolerance and complement the limited influence of particular pollen [149]. Nevertheless, a monofloral diet may occasionally reduce the mortality of honeybees, compared with that of bees denied pollen [150]. Nutritional stress associated with the monoculture and consumption of monofloral pollen affected bee gut microbiota and immunity and increased pathogen infection, such as *N. ceranae*. Under laboratory conditions, honeybees fed with the monofloral pollen of *Eucalyptus grandis* exhibited less gut microbiota and reduced gene immunity compared with those of honeybees fed with multifloral pollen [151].

Honeybee food supplements can boost the health of colonies and minimize the impact of stressors during pollen foraging unavailable periods [152]. Artificial food supplements using fresh and dry Cyanobacteria, *Arthrospira platensis*, exhibited a marked overlap in proteome expression patterns and similar dietary protein absorption, similar to natural pollen [153]. *Chlorella sorokiniana* played a role in bees development via increased fat deposition, Vg transcript levels, decreased target of rapamycin, and InR2 transcript levels [154].

Other marine organisms such as seaweeds used in the production of Hive Alive reduced *N. ceranae* spores and increased the number of workers [155]. When bacteria-like bifidobacteria and lactobacilli were added to sugar syrup, the area of pollen storage and harvestable honey dramatically increased. It promoted the growth of Acetobacteraceae and Bifidobacterium spp, which are important for bees nutrition [156].

The role of dietary phytochemicals in honeybees’ sensitivity to pesticide exposure cannot be overstated. The dietary supplementation with *p*-coumaric and indole-3-acetic acids (20 μM) enhanced the survival of bees exposed to tau-fluvalinate (~20%) [157]. Abscisic acid (ABA), a natural component found in nectar, honey, pollen, and honeybees, is critical for bees’ health. In the dietary strategies tested, ABA supplementation positively impacted the population dynamics of *A. mellifera* colonies during overwintering and nosemosis at the colony level (prevalence) [158]. Most protein supplements are made from high-protein foods such as soy, yeast, or natural pollen, such as MegaBee^®^, Feed-Bee^®^, Bee-Pro^®^, Global Patties^®^, and Ultra Bee^®^ [159]. Non-protein amino acids such as gamma-aminobutyric acid and beta-alanine were used to reduce the possibility of beneficial microbiota imbalance following antivarroa and antinosemosis therapy [160]. Artificial protein supplements such as *Beewell Aminoplus* boosted bee survivability, decreased Nosema spore levels, had an immunostimulatory impact, and increased the antioxidative protection of the bees against *N. ceranae* [161,162]. Another study estimated that colonies fed on pollen had lower titers of BQCV and Nosema compared with colonies fed on protein supplements, such as Bee-ProV [163]. In addition, FeedBeeVR did not affect queen development compared with a sugar-based diet supplemented with honey and fresh pollen [164]. The supplementation of honeybees with inadequately chosen probiotics or both probiotics and prebiotics cannot prevent the development of Nosemosis but can deregulate insect immune systems, which may result in considerable bee mortality [165].

### 5.2. Natural Products as Alternative Sources

The high cost of chemical treatments and the risk of toxic residues presence in bee products pose problems [166]. The existing treatments may only kill some mites, leaving them more resistant to breed in the next generation, thereby increasing their resistance over time [167]. As a result, scientists worldwide are working to develop effective medicines that have minimal side effects on bee products.

Natural products are one of the sources of these treatments. Essential oils such as thymol, linalool, and camphor, as well as cocktails of thymol, eucalyptol, menthol, and others, have been confirmed to be particularly efficient in suppressing Varroa mites. These types of essential oils were discovered to lower mortality rates among bees in diseased colonies [168]. Although natural products therapy has fewer side effects than chemical therapy, the efficacy of these substances varies depending on the climate and colony condition [169].

Recently, Chinese herbal medicine has demonstrated a unique antiviral effect for both human and animal life. Honeybees are at risk from SBV and Chinese sacbrood virus (CSBV). Infected larvae will not develop into pupae and will eventually die, and there is currently no effective cure for the virus [20]. *Radix isatidis*, a Chinese herbal remedy, was primarily utilized to treat human influenza viruses. It has recently been proved to effectively regulate CSBV by suppressing its replication, increasing immunological response, and extending the lifespan of CSBV infection larvae, thereby lowering death rates and preventing CCD [170]. DWV and Lake Sinai virus are two RNA viruses with positive strands that kill honeybees. Bees fed with polypore mushroom extracts exhibited a strong ability to diminish both virus larvae. Modified porphyrins, which are mostly produced by living organisms, can reduce spore burdens in bees and increase the survival likelihood of bees infected with RNA viruses [171].

Essential oils and extracts of tea tree oil NPs exhibited an extremely high antibacterial effect against AFB and European foulbrood (EFB) *in vivo*. They inhibited the growth of AFB and EFB at low concentrations; this had a high effect on the survival of the colony [172]. Nosemosis is one of the most widespread diseases among honeybee colonies. It is mainly caused by the fungal microsporidian parasites *N. apis* and *N. ceranae*, which can cause honeybee losses. The anti-fungal agent, fumagillin, is the most common treatment for nosemosis. However, the causative agents may acquire treatment resistance, which could pose a threat in the future [173]. Natural products are used as an alternative treatment. *Cryptocarya alba* essential oils and monoterpenes have been demonstrated to possess potent anti-fungal properties against Nosemosis. They can slow and stop the spread of disease, thereby increasing the survival chances of the colony [174]. *N. ceranae* was treated with ApiHerbfi, a nutritional supplement, and Api-Bioxalfi, a veterinary drug, and the infections were reduced substantially [175]. Furthermore, glucosinolates derived of *Brassica nigra* and *Eruca sativa* were used, and hence *N. ceranae* was inhibited in an *in vitro* model [176]. However, in the field trials, the treatments had no significant effects on colonies development or bees mortality when compared to the negative controls, and the treatments had no effect on the prevalence of *N. ceranae* infected bees in both pre- and post-treatment samples, so further optimization of the dose or application methods is needed [177].

### 5.3. Nanomaterials as Novel Alternative Approaches

Nanotechnology is one of the most active areas in research that has proven to be extremely versatile and has sparked a revolution in medical treatments, fast diagnoses, cellular regeneration, and medication delivery [178,179]. It has been employed to discover novel therapies for honeybee diseases, as current antibiotics do not entirely eradicate the infection. Antibiotic use over time results in an accumulation of antibiotics in honey, which can be hazardous to human health [180]. Furthermore, it is damaging to the gut microbiota of bees, impairing their metabolism, which results in weak immunity, high disease risk, and short bee lifespans [181]. As a result, through ultraviolet–visible (UV–VIS) spectrophotometry and scanning electron microscopy, researchers discovered that camphor tree silver NPs could positively regulate the isolated bacterial pathogens of AFB and EFB [182]. Similarly, tea tree oil NPs have been utilized in other studies to treat or prevent AFB and EFB due to their high efficacy against Paenibacillus sp. and *M. plutonius* strains, respectively, which are the causal agents [172]. AgNPs have anti-microsporidian activity and could be effective components of formulations for treating or preventing microsporidia *Nosema bombycis* [183]. However, additional research is needed on this topic in order to develop nanoparticle treatments that will reduce bee diseases while having no negative impact on honeybee health.

### 5.4. Organizations and Initiatives Directed to Saving the Bees

Organizations and initiatives are directed to save the bees in response to honeybee colony losses. Working groups such as COLOSS, a COST initiative funded by the EU Science Foundation, have been formed to address the global loss of managed honeybee colonies. By offering strategies to reduce the risk of this problem, COLOSS is playing an important role in identifying and mitigating colony losses [184]. One of these strategies was adequate nutrition that was a key factor for honeybees’ growth and colony development. Several studies show that the nutritional quality of diet is directly proportional to the ability of bees to face challenges or stressors [185]. However, diverse feeding regimes are used by beekeepers all throughout the world, particularly during the winter season, and these fluctuate from nation to nation and beekeeper to beekeeper. For example, in China, which has a large number of managed honeybee colonies, beekeepers feed winter bees with honeycombs or sugar [186]. In Egypt, colonies are supplied with artificial diet cake and syrup supplemented with vitamin C; this combination has a significant impact on colonies’ growth compared to sugar syrup only [187]. The feeding regime in Ethiopia is honey and water and shiro (roasted spiced pulses flour) to overcome the feed shortage in the dry season [188]. Apiary management in most nations necessitates an initial feeding of carbohydrate and protein. Protein is not always used; in the spring, a carbohydrate such as sucrose syrup or high corn syrup can be used to replicate nectar flow and enhance brood raising. This extra feed is utilized in the fall to increase fructose storage for the winter [189]. As a result, frequent COLOSS nutrition task force meetings could serve to protect bees from varied stressors, particularly during the winter seasons, by exchanging different apiary management protocols and feeding regimes used in different regions of the world.

## 6. Conclusions

Honeybees are important pollinators for humans and ecosystems. Unfortunately, CCD, a serious threat to the beekeeping industry, has recently been reported worldwide. It is caused by a variety of stressors that affect the immune system of bees, such as pathogens, insecticides, and inadequate diets. Scholars and governments have universally agreed that there is no single cause is to be blamed and that the causes are interconnected. However, further research is required to understand the mechanisms behind the interactions among different stressors and to discover more important genes and signaling pathways involved in honeybee stress responses. First, beekeepers must consider these aspects by planting floral-rich vegetation around the apiary and using proper dietary supplements. Second, experts could agree on a scientific plan for the treatment and management of the related diseases and pests, including the development of new nanotechnology-based remedies. Finally, organizations and stakeholders should pay attention to training to improve the efficiency of breeders and recent graduates in apiary management.

## Figures and Tables

**Figure 1 vetsci-09-00199-f001:**
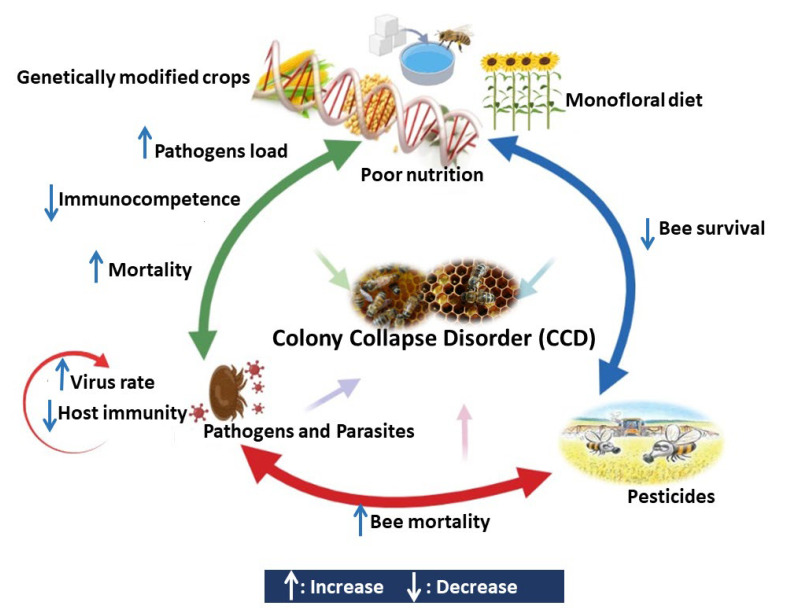
The impact on honeybees’ health when exposed to interaction between environmental and ecological stressors.

**Figure 2 vetsci-09-00199-f002:**
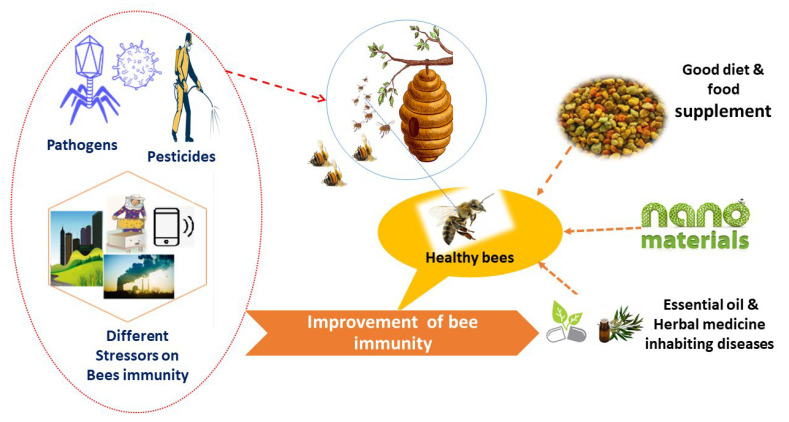
Factors that can improve honeybees immunity against different stressors.

## Data Availability

Not applicable.

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
