# Peer review of "Bee Stressors from an Immunological Perspective and Strategies to Improve Bee Health"

_vetsci, 2022, doi:10.3390/vetsci9050199_

Round 1

Reviewer 1 Report

Major revision:

  1. The name of CCD was from colony sudden disappear. In fact, Cox-Foster et al. had been reported that Israeli acute paralysis virus (IAPV) was the cause of induced the colony collapse. Although subsequent studies can not confirm the link between IAPV and CCD, IAPV was indeed one of the reasons for responsible to colony decline. In addition, recent studies showed that IAPV is not only infect honey bee, but also bumblebee, stingless bee, hornet and even ant. Therefore, IAPV as one of bee viruses should be described in detail. On the contrary, the authors did not mention a word about the relations between IAPV and colony decline except IAPV induced immune response.  
  2. The author should separate the immune response and pathogenesis in a good way, but not mix them in a paragraph.
  3. The title is “Immunity and Honey Bee Health”, but it is stated all of pathogens and interaction among pathogens and abiotics. In my opinion, it should describe pathogens, parasites or other stressors harm to immune system, and progress to impact the health of honey bee, and then take some strategies to improve the health. The authors spend a lot of time to describe the pathogens impacted CCD but not immunity.

Minor revision:

  1. The Figure resolution has to be improved.
  2. The humeral responses should be humoral responses.
  3. The language should be improved by native speaker.

Author Response

  1. The name of CCD was from colony sudden disappear. In fact, Cox-Foster et al. had been reported that Israeli acute paralysis virus (IAPV) was the cause of induced the colony collapse. Although subsequent studies can not confirm the link between IAPV and CCD, IAPV was indeed one of the reasons for responsible to colony decline. In addition, recent studies showed that IAPV is not only infect honey bee, but also bumblebee, stingless bee, hornet and even ant. Therefore, IAPV as one of bee viruses should be described in detail. On the contrary, the authors did not mention a word about the relations between IAPV and colony decline except IAPV induced immune response.

Response: Adjusted and yellow highlighted in the revised version

Lines 215-218: “Israeli acute paralysis virus (IAPV) has been identified as a major cause of colony collapse disorder (CCD) [60], although other studies did not directly associate CCD and the presence of IAPV, successive replication of IAPV was shown to affect the colonies health and thus possibly determine their survival [61–63]. In addition, IAPV is not only  source of infection to honey bees, but also bumblebees [64], stingless bees [65], hornets [66] and even ants [67]..”

  1. The author should separate the immune response and pathogenesis in a good way, but not mix them in a paragraph.

Response: We modified and rearranged the paragraphs accordingly.

  1. The title is “Immunity and Honey Bee Health”, but it is stated all of pathogens and interaction among pathogens and abiotics. In my opinion, it should describe pathogens, parasites or other stressors harm to immune system, and progress to impact the health of honey bee, and then take some strategies to improve the health. The authors spend a lot of time to describe the pathogens impacted CCD but not immunity.

Response:  The title changed to “Bee Stressors from an Immunological Perspective and Strategies to Improve Bee Health”

  1. Minor revision:

1. The Figure resolution has to be improved.

Response: Adjusted

2. The humeral responses should be humoral responses.

Response: Adjusted

3. The language should be improved by native speaker

Response: The authors read the manuscript thoroughly and amend it accordingly. The manuscript was subjected to English editing.

Reviewer 2 Report

The article is very interesting and the organizational scheme of the manuscript is very clear and appealing. However, some parts are developed superficially because there are so many topics.

Many bibliographical references are missing. In the reviews every sentence needs a bibliographical reference. Here there are whole paragraphs where references are absent. Please add a bibliographical reference in each sentence.

Scientific names of plants (e.g. Eucalyptus grandis, etc.), animals (e.g. Varroa destructor, Apis mellifera, etc.), fungi (e.g. Nosema ceranae, Ascosphera apis, etc.), bacteria (e.g. Arthrospira platensis, Paenibacillus, M. plutonius), and Latin words (e.g. viceversa, in vivo), also in abbreviated forms, must be written in italics. Please correct throughout the text and tables.

The two tables enrich the work a lot, but I suggest reorganizing them internally, perhaps by grouping the first column in groups of the same stressor to make them easier to read. Furthermore, if there is no order of stressors, but the bees are subjected to them simultaneously, I suggest putting the same stressors in the same column.

it is necessary to revise the English because some sentences are unclear, especially in chapters 1 and 2.

Chapter 2 does not mention glucose oxidase, which is a very important and well-studied enzyme secreted by bees. It is important to add it.

Author Response

The article is very interesting and the organizational scheme of the manuscript is very clear and appealing. However, some parts are developed superficially because there are so many topics.

We would like to thank the reviewer for the time spent on reviewing our manuscript and the comments that helped us improving the article.

  1. Many bibliographical references are missing. In the reviews every sentence needs a bibliographical reference. Here there are whole paragraphs where references are absent. Please add a bibliographical reference in each sentence.

Response: Adjusted

  1. Scientific names of plants (e.g. Eucalyptus grandis, etc.), animals (e.g. Varroa destructorApis mellifera, etc.), fungi (e.g. Nosema ceranaeAscosphera apis, etc.), bacteria (e.g. Arthrospira platensis, Paenibacillus,  plutonius), and Latin words (e.g. viceversain vivo), also in abbreviated forms, must be written in italics. Please correct throughout the text and tables.

Response: Adjusted

  1. The two tables enrich the work a lot, but I suggest reorganizing them internally, perhaps by grouping the first column in groups of the same stressor to make them easier to read. Furthermore, if there is no order of stressors, but the bees are subjected to them simultaneously, I suggest putting the same stressors in the same column.

Response: Adjusted

  1. it is necessary to revise the English because some sentences are unclear, especially in chapters 1 and 2.

Response: The authors read the manuscript thoroughly and amend it accordingly. The manuscript was subjected to English editing.

  1. Chapter 2 does not mention glucose oxidase, which is a very important and well-studied enzyme secreted by bees. It is important to add it.

Response: Lines 133-137: “Glucose oxidase (GOX) is an antiseptic enzyme found in nectar and larval diet, and an additive to prolong the products shelf life. In the hypopharyngeal glands, GOX is a catalytic enzyme that catalyzes the conversion of β- d- glucose to gluconic acid and hydrogen peroxide (H2O2) [26] and provides the social immunity. H2O2 functions as an antiseptic, preventing pathogen growth in honeybee larval diet [27].”

Round 2

Reviewer 1 Report

It can be published.

Author Response

Comments and Suggestions for Authors

It can be published.

Response:  We would like to thank the reviewer for the time spent on reviewing our manuscript and the comments that helped us improving the article.